# Instant Cascara Beverage as a Neuroimmune Modulator of the Brain–Gut Axis: Sex-Dependent Effects in Healthy Rats

**DOI:** 10.3390/ijms262110691

**Published:** 2025-11-03

**Authors:** Paula Gallego-Barceló, Yolanda López-Tofiño, Laura López-Gómez, Gema Vera, Ana Bagues, Jesús Esteban-Hernández, María Dolores del Castillo, José Antonio Uranga, Raquel Abalo

**Affiliations:** 1Department of Basic Health Sciences, Faculty of Health Sciences, University Rey Juan Carlos (URJC), 28922 Alcorcón, Spain; paula.gallego@urjc.es (P.G.-B.); yolanda.lopez@urjc.es (Y.L.-T.); laura.lopez.gomez@urjc.es (L.L.-G.); gema.vera@urjc.es (G.V.); ana.bagues@urjc.es (A.B.); jose.uranga@urjc.es (J.A.U.); 2High Performance Research Group in Physiopathology and Pharmacology of the Digestive System (NeuGut-URJC), University Rey Juan Carlos (URJC), 28922 Alcorcón, Spain; mdolores.delcastillo@csic.es; 3Working Group of Basic Sciences on Neuropathic Pain of the Spanish Pain Society, 28046 Madrid, Spain; 4Working Group of Basic Sciences on Cannabinoids of the Spanish Pain Society, 28046 Madrid, Spain; 5Associated I+D+i Unit to the Institute of Medicinal Chemistry (IQM), Scientific Research Superior Council (CSIC), 28006 Madrid, Spain; 6High Performance Research Group in Experimental Pharmacology (PHARMAKOM-URJC), University Rey Juan Carlos (URJC), 28922 Alcorcón, Spain; 7Department of Medical Specialties and Public Health, Area of Preventive Medicine and Public Health, Faculty of Health Sciences, University Rey Juan Carlos (URJC), 28922 Alcorcón, Spain; jesus.esteban@urjc.es; 8High-Performance Research Group on Environmental Risks for Health and the Environment, University Rey Juan Carlos (URJC), 28922 Alcorcón, Spain; 9Food Bioscience Group, Department of Bioactivity and Food Analysis, Instituto de Investigación en Ciencias de la Alimentación (CIAL) (CSIC-UAM), Calle Nicolás Cabrera 9, 28049 Madrid, Spain; 10Working Group of Basic Sciences on Pain and Analgesia of the Spanish Pain Society, 28046 Madrid, Spain

**Keywords:** coffee by-product, enteric nervous system, brain–gut axis, immunity, phytochemicals, visceral pain

## Abstract

Instant Cascara (IC), a beverage obtained from dried coffee cherry pulp, represents a sustainable hydration option rich in bioactive phytochemicals, such as caffeine, chlorogenic acids, and melanoidins, which may provide effects beyond basic nutrition. This study evaluated the impact of three weeks of IC consumption on somatic and visceral sensitivity, and on neural and immune markers in the colon of male and female healthy Wistar rats. Behavioral tests showed that IC increased locomotor activity and somatic sensitivity in females (*p* < 0.05). Although control females were more sensitive to visceral pain than males (*p* < 0.05), IC intake did not significantly alter pain sensitivity in either sex. Histological and immunohistochemical analyses in the colonic myenteric plexus revealed higher enteric glial cell density and glia-to-neuron ratio (*p* < 0.01), but lower calcitonin gene-related peptide (CGRP)-positive fiber density (*p* < 0.001) in IC-treated compared to control females. Macrophages decreased in IC-treated compared with control males in the colon wall (*p* < 0.05), whereas their number increased in IC-treated females compared to IC-treated males (*p* > 0.0001). Visceral pain responses are associated with complex sex-dependent neuroimmune changes in the colon. Interestingly, IC effects appear mild under healthy conditions, possibly due to compensatory mechanisms exerted by its different phytochemicals. Further investigation is needed to determine the effects of IC in pathological situations involving visceral hypersensitivity, such as brain–gut axis disorders.

## 1. Introduction

Instant Cascara (IC) is a novel beverage derived from the dried pulp of the coffee cherry, an agricultural by-product usually discarded during coffee processing. Its development represents a strategy aligned with sustainability and the circular economy while also responding to the growing demand for functional beverages, where consumers increasingly value natural products with added health benefits [1]. Dried coffee cherry pulp was authorized by the European Commission as a novel food (Commission Implementing Regulation (EU) 2022/47), under Regulation (EU) 2015/2283 on novel foods, confirming its safety for human consumption and permitting its commercialization within the European Union [2]. Compared with traditional coffee husk infusions, IC is obtained combining hot water extraction and drying processes under control conditions to preserve and concentrate the natural bioactive compounds of the dried coffee cherry pulp, including caffeine, chlorogenic acids, melanoidins, polysaccharides, and free amino acids such as gamma-aminobutyric acid (GABA) [3,4]. Consequently, IC provides both nutritional and sensory properties together with the potential for synergistic physiological effects, which clearly distinguishes it from other coffee by-product beverages [3,4].

IC contains bioactive compounds with well-documented physiological activities, but the complexity of the food matrix may modulate their overall effects. Caffeine acts through antagonism of adenosine receptors, enhancing neurotransmission and modulating the release of mediators such as calcitonin gene-related peptide (CGRP) and substance P [5,6]. Polyphenols, particularly chlorogenic acids, display antioxidant, anti-inflammatory, and neuroprotective activities that regulate immune cell activation and oxidative stress [7,8,9]. Melanoidins and polysaccharides behave as dietary fibers with prebiotic activity, supporting microbial fermentation and the production of short-chain fatty acids (SCFAs), which contribute to intestinal barrier function and immune regulation [10]. GABA may influence enteric neurotransmission and glial reactivity by reducing afferent excitability [11]. Together, these bioactive compounds may exert neuroimmune modulatory effects in the whole body, in general, and in the gastrointestinal tract in particular.

In the colon, the interplay between enteric neurons, glial cells, and mucosal immune cells sustains local homeostasis [12,13,14,15]. When this balance is disrupted, afferent neurons become sensitized, lowering their activation threshold and generating visceral hypersensitivity, which is considered the physiological basis of abdominal pain in different pathological conditions [16,17]. Furthermore, because intestinal neuroimmune modulation is a central element of the brain–gut axis (BGA), alterations at the colonic level may have consequences on central processing of pain. Afferent neurons transmit nociceptive signals to the central nervous system (CNS), where their processing is further shaped by descending pathways influenced by stress, emotional state, and prior experiences [13,18]. Biological sex is also a key factor in this modulation, as differences in immune responses, hormonal regulation, and central pain processing increase susceptibility to neuroimmune imbalance [19,20]. Clinically, dysregulation of the BGA is associated not only with visceral disorders such as irritable bowel syndrome but also with extraintestinal manifestations including fatigue and musculoskeletal pain typical of fibromyalgia, which often coexist through shared mechanisms of spinal convergence and central sensitization [19,20,21]. These manifestations are more frequent and severe in women, reflecting sex-related biological differences [19,20]. In this context, IC—through its combined effects on immunity, glial activity, and CGRP release—may influence not only visceral sensitivity but also somatic pain, through shared mechanisms of central sensitization [17,22].

Previous animal studies have shown that regular IC consumption is safe and does not significantly alter gastrointestinal motility in vivo [23], whereas subtle sex-dependent effects in colonic contractility were detected in vitro [24]. The aim of this study was to further characterize the effects of IC on the BGA, namely through the analysis of its impact on visceral sensitivity, and on key components of the colonic neuroimmune axis such as enteric glial cells, CGRP-positive fibers, and immune cell infiltration in the colonic wall. Somatic sensitivity was evaluated for comparison, and locomotor activity was also assessed. Importantly, the possible influence of sex dimorphism on IC effects were determined.

## 2. Results

### 2.1. General Health and Behavioral Studies

#### 2.1.1. General Health Parameters

General health parameters (body weight, food, and liquid intake) were monitored weekly throughout the three-week study of IC effects in both male and female healthy rats in cohort 1. Over this period, male control rats gained 35% of their baseline weight, while female controls showed no weight change (Appendix A). A significant sex difference in body weight was evident across the study, with control males consuming more food (23 g/rat/day) and liquid (33 mL/rat/day) than females (16 g/rat/day and 29 mL/rat/day, respectively), the difference being statistically significant for food intake (Appendix A). IC exposure did not significantly alter body weight (Appendix A), food consumption (Appendix A), or liquid intake (Appendix A) compared to the respective control groups.

#### 2.1.2. Locomotor Activity and Somatic Sensitivity

During the second week of the study, locomotor activity was assessed using an actimeter in animals from cohort 1. Over time, there was a statistically significant decrease in the number of photobeam crosses in all experimental groups (*p* < 0.05) (Table 1), but at all time-intervals, control males and females presented a similar number of photobeam crosses (*p* > 0.05). When comparing the IC exposed groups to their corresponding controls, IC did not modify locomotor activity in males, but a statistically significant increase was observed during the first period (0–10 min) in females (*p* < 0.05) (Table 1).

On the same day, somatic sensitivity was evaluated in these animals using two complementary tests: the von Frey test to measure responses to non-noxious tactile stimuli, and the pressure administered measurement (PAM) test to assess skeletal muscle responses to noxious pressure [25].

In the von Frey test (Figure 1A), control males and control females had a similar threshold, with no statistically significant differences between them. In the case of animals exposed to IC beverage, both sexes tended to exhibit an increased threshold, suggesting a slight reduction in tactile sensitivity, although the differences were not statistically significant when compared to their control groups.

In the PAM test (Figure 1B), control females tended to display a higher threshold compared to control males, without the differences reaching statistical significance (*p* = 0.1205). Males administered with the IC beverage also showed a slight, non-statistically significant increase in this parameter compared to their controls (*p* > 0.05). In contrast, IC females showed a statistically significant reduction in the nociceptive threshold compared to control females, suggesting IC increased skeletal muscle sensitivity in this sex (*p* < 0.05).

Finally, the estrous cycle was evaluated to check for its possible influence on the behavioral effects of IC in females. In this study, most females were in proestrus/estrus (66% in controls, 78% in IC), with no significant differences in phase distribution between groups (Appendix A).

#### 2.1.3. Visceral Sensitivity

Visceral pain was assessed during the third week of the study, using the colorectal distension test and a phasic stimulation protocol, which causes sex-dimorphic abdominal pain responses in healthy rats, with females displaying higher sensitivity than males [26].

The number of contractions per minute in the control male and female groups progressively increased with intracolonic pressure, with a maximum of 10 and 20 contractions per minute at 80 mmHg, respectively, suggesting a stronger response in females than in males (Figure 2A), and IC slightly increased the response in both sexes (Figure 2A).

The duration of contractions (Figure 2B) was around 1–2 s at all pressures in all experimental groups, with no noticeable differences among them, except for the response observed at the initial 0 mmHg pressure, where females exhibited contractions while males did not.

Finally, the percentage of time contracting the abdomen (Figure 2C), which is a representation that provides combined information on both the number of contractions and the mean contraction duration, showed a very similar profile as the graph shown in Figure 2A. Thus, an increase was observed at 20–80 mmHg in females compared to males, irrespective of the treatment received (water or IC), and regular IC consumption further increased the percentage of time in contraction in males, and more so in females (Figure 2C).

Statistical analysis supported these findings, revealing a strong main effect of pressure (≈30–58% of total variance, *p* < 0.0001) and a significant influence of sex (≈5–18%, *p* ≤ 0.01), with females showing greater contractile activity. Beverage effects were minor (≈1–2%), although a significant pressure × beverage interaction was detected for the percentage of time in contraction (*p* = 0.0245), suggesting a mild, pressure-dependent modulation by IC. Additional pressure × sex interactions (≈4–6%) reflected sex-related differences in pressure–response profiles, while no significant sex × beverage or three-way interactions were found. Post hoc comparisons confirmed that these effects were mainly driven by sex rather than beverage intake.

In the estrous cycle analysis performed after the visceral pain study was ended, most females were in proestrus/estrus (89% in both control and IC groups), with no significant differences in phase distribution between groups (Appendix A).

### 2.2. Histological and Immunohistochemical Colonic Studies

#### 2.2.1. Colonic Myenteric Plexus

Whole-mount preparations of distal colon from cohort 2 animals were used to evaluate different components of the myenteric plexus, the subdivision of the enteric nervous system that intrinsically controls gastrointestinal motility and also contributes to visceral sensitivity [27]. Enteric glial cells were identified by Sox-10 immunolabelling (Figure 3A) [28], while CGRP-immunoreactive (IR) fibers (Figure 3B) were analyzed as markers of sensory innervation [29]. For glial cells, both their absolute density and their ratio relative to HuC/D-positive myenteric neurons were quantified, since this ratio has been reported to vary under pathological conditions such as intestinal inflammation [30].

A statistically significant increase (*p* < 0.01) of the total density of Sox-10 IR glial cells was observed in IC females compared with control ones (Figure 3B). Furthermore, the ratio of Sox-10-IR enteric glial cells per myenteric neuron also showed statistically significant differences (*p* < 0.01), with higher ratios observed in IC females compared to control females (Figure 3C). No statistically significant differences were observed for the packing density of glial cells and extraganglionic glial density based on sex or beverage. 

Compared with control males, in control females, there was an increase in the density in CGRP-IR varicose fiber density (measured semiquantitatively) but the difference was not statistically significant (*p* = 0.06, Figure 3E). However, in IC-exposed females, CGRP-IR varicose fiber density was significantly reduced compared to control females (*p* < 0.01).

#### 2.2.2. Colonic Wall Immunocytes

Histological sections of the distal colon wall from cohort 2 animals were analyzed to assess immune cell infiltration. Specific staining/immunohistochemical methods were applied to identify different immune cell populations, including mast cells, neutrophils, and macrophages (see Figure 4 for representative images).

No eosinophil infiltration or signs of eosinophilia were observed in any experimental group. Mast cell counts were generally low in all groups, without statistically significant differences among experimental groups (Figure 4B). Neutrophil quantification showed a slight increase in control females compared to control males, but this difference was also not statistically significant (Figure 4D). Additionally, IC treatment led to a modest increase in neutrophil numbers in both sexes; however, the changes did not reach statistical significance compared to sex-matched controls.

In contrast, macrophage analysis revealed statistically significant effects of IC consumption (Figure 4F). While M2 macrophage counts were similar between control males and females, IC treatment significantly decreased M2 macrophages in males compared to controls (*p* < 0.01), whereas in females, IC exposure led to a significant increase in macrophage numbers compared to IC-treated males (*p* < 0.0001), although the difference with control females did not reach statistical significance.

In these animals, vaginal cytology was performed immediately before sacrifice to determine estrous cycle phases, and no statistically significant differences were found between control and IC groups (Appendix A).

## 3. Discussion

In this study, the effects of regular IC consumption, a beverage developed from dried coffee cherry, were evaluated on somatic and colonic pain, as well as on neural and immune markers in the colonic wall of healthy rats of both sexes. Interestingly, some sex-specific effects of IC were observed (Table 2).

The differences between control males and females in body weight gain and food and liquid intakes followed the expected pattern for this rodent species, whereas no statistically significant difference was detected in these parameters associated with the regular consumption of IC in either sex. This agrees with previous studies reporting the limited impact of coffee and related products on these parameters in experimental animals [10,23,24,31].

In contrast, some sex-dependent effects of IC were observed in the behavioral studies used to evaluate somatic and visceral pain thresholds and the histological and immunohistological studies evaluating the neuroimmune axis. Compared with control animals, IC induced in females a significant increase in locomotor activity and musculoskeletal sensitivity, while visceral responses showed only a mild, non-significant enhancement. Mixed-model analyses, which accounted for repeated measures and within-subject variability, confirmed that visceral contractile activity increased proportionally with pressure and was primarily influenced by sex, with females showing stronger responses than males. Treatment effects were modest and pressure-dependent, indicating that IC produced only a slight modulation of visceral reflexes. The lack of a statistically significant effect on visceral pain, despite the slight increase seen in IC-treated females, could be due to compensatory mechanisms in the colonic innervation. Visceral pain is a complex process that depends on the interaction of many factors—nervous, glial, and immune—which can balance each other and keep sensitivity within normal limits [32]. Indeed, behavioral changes associated with IC intake were accompanied by a higher enteric glial cell density and glia-to-neuron ratio, but by fewer CGRP-positive fibers in the myenteric plexus. Regarding immunocytes infiltrating the colon, only macrophages showed significant changes, suggesting a selective rather than general immune effect. Their contribution could help explain the slight rise in visceral sensitivity in females, although compensatory neuroimmune mechanisms probably prevented clear functional differences. Such adjustments might become more relevant under pathological conditions, when this balance is altered and inflammatory mediators such as tumor necrosis factor alpha (TNF-α) and interleukin-6 (IL-6) could play a role [33]. In contrast, the reduction in anti-inflammatory macrophages in males did not translate into higher visceral sensitivity, likely because other compensatory elements helped preserve colonic function. These may include preserved CGRP-positive fibers or stable glial density that maintained normal sensitivity.

Whatever the case may be, the lack of statistically significant differences in the estrous cycle phase distribution between control and IC females at the different time points where these studies were performed (Appendix A), suggests that the observed IC effects cannot be attributed to intra-group variability related to estrous cycle phase. Most likely, instead, the sex-specific behavioral and neuroimmune effects of IC are due to the sex-dependent effects of the bioactive phytochemicals composing the beverage.

Among the compounds present in IC, caffeine is the most extensively studied in beverages, owing to its psychostimulant and analgesia-modulating actions. In our study, caffein intake was 15.9 mg/kg/day in males and 18.6 mg/kg/day in females, corresponding to a human equivalent dose of approximately 2–3 cups of coffee per day [34,35]. Caffeine acts mainly as a non-selective antagonist of adenosine receptors (A1, A2A, A2B, A3) and as an agonist of ryanodine receptors (RyR), showing an inverted-U effect on locomotion: stimulation at low-to-moderate doses and inhibition at higher ones [36]. Importantly, A1 activation is analgesic in inflammatory and neuropathic pain, whereas its blockade may enhance visceral sensitivity [37,38]. In the gut, adenosine receptors regulate motility and sensitivity: A1/A3 are found in myenteric neurons, A2A in cholinergic terminals, and A2B in glial cells [39]. Their antagonism by caffeine enhances excitatory neurotransmitter release, while stress-related adenosine accumulation can shift signaling to A2A/A3, promoting colonic contractility and discomfort [40,41,42]. Consistent with this, our intracolonic balloon distension protocol, using established experimental ranges (10–80 mmHg) [43,44], showed that IC-exposed females tended to exhibit increased responses, although these differences did not reach statistical significance at any pressure tested. This effect, despite not reaching statistical significance in the present study, could facilitate colonic hyperalgesia (rather than allodynia) and abdominal discomfort, likely linked to neurotransmitter and glial changes. This would be in agreement with our previous ex vivo finding of enhanced non-muscarinic contractile colonic muscle responses to high-frequency stimulation in females [24], which may mimic the impact of high-pressure colorectal distension. Regular caffeine consumption can also induce adaptive changes in A1 receptor expression and function in the CNS and enteric nervous system (ENS) [45], which may underlie the increased enteric glial density and reduced CGRP-IR fibers found in IC females. These changes may involve A2B-mediated signaling under high intracolonic mechanical stimulation.

The sex-specific effects of IC may be explained by both pharmacokinetic factors affecting caffeine differentially in males and females, such as its slower metabolism or its greater adipose accumulation in females compared to males [46,47]. Likewise, pharmacodynamic factors may also be influential, since estrogens modulate adenosine and dopamine receptor function and enhance RyR-mediated Ca^2+^ release [48,49]. Indeed, females in proestrus/estrus are more sensitive to somatic and visceral pain stimuli [45,50,51]. Although cycle distribution was similar between our two female groups when the two main categories of estrous phases were considered, suggesting a minor influence of this factor, it cannot be discarded that part of the differential effect observed here could be due to intragroup differences among the four phases (Appendix A).

In any case, IC is a complex food matrix, and its effects may not be dependent only on caffeine, as shown for other food matrices. For example, locomotion was unchanged with 80 mg/kg/day pure caffeine but increased with cola [31], and was higher with green tea (~69 mg/kg/day caffeine) than with pure caffeine (100 mg/kg/day) in female mice, likely due to additional compounds such as polyphenols [52].

In particular, chlorogenic acid (CGA), the main phenolic compound in IC, is recognized for its antioxidant and anti-inflammatory effects [53]. Total phenolic intake was estimated at 91.7 mg/kg/day in males and 112 mg/kg/day in females, with CGA contributing 125–147 mg/kg/day in males and 146–172 mg/kg/day in females, exceeding the doses typically used in animal studies. One study observed that chronic voluntary intake of caffeinated and decaffeinated coffee (providing ~10–20 mg/kg/day CGA) improved the antioxidant status in mice of both sexes by reducing lipid and protein oxidation and increasing glutathione levels and glutathione peroxidase activity in the frontal cortex, a brain region involved in the regulation of stress responses and pain processing [45]. However, behavioural changes (increased locomotion, coordination, and motivation) were only seen in females consuming caffeinated coffee (with an estimated intake of ~1.5 mg/day caffeine, corresponding to ~30 mg/kg/day), suggesting that caffeine was responsible for these behavioural effects, while CGA mainly produced antioxidant actions. Although further studies are needed to evaluate the extent to which CGA content in IC caused similar protective effects on the brain of our animals and whether the ENS was equally protected, IC has already demonstrated promising antioxidant and anti-inflammatory effects in cell cultures [3].

GABA is the main inhibitory neurotransmitter in the CNS and exerts its action through ionotropic GABA_A_ and metabotropic GABA_B_ receptors. GABAergic modulation of pain is well documented at the spinal level, through the activation of GABA_A_ and GABA_B_ receptors [54]. In the colon, GABA promotes circular muscle relaxation through its actions on GABA_A_ receptors present in motor neurons releasing inhibitory neurotransmitters (vasoactive intestinal peptide, nitric oxide, ATP …), whereas it may increase or decrease acetylcholine release from excitatory motor neurons by acting on GABA_A_ or GABA_B_, leading to increased or decreased contractility, respectively [55]. Thus, the GABAergic system is a relevant neuromodulatory system in the ENS, balancing excitatory and inhibitory neural signals.

In addition, the GABAergic system is considered a “connecting bridge”, ensuring a functional cooperation between the nervous system and the immune system. Whereas in physiological conditions, GABA could contribute to maintain the “tolerogenic state” to harmless antigens in the enteric microenvironment, including food antigens and commensal microbiota components, under inflammatory conditions it exerts anti-inflammatory actions via inhibition of inflammatory events mediated by different immune cells, including macrophages, through both GABA_A_ and GABA_B_ receptors. Importantly, no frank inflammatory damage was observed in the colonic wall associated with IC exposure in either male or female rats [24]. Furthermore, there was no eosinophil infiltration, a typical feature of chronic microinflammatory conditions [56]. However, IC beverage affected macrophages differently in males and females. In males, IC beverage reduced anti-inflammatory macrophages, while in females, with stronger responses to colorectal distension, it increased them. Although in our study, the IC beverage provided only low amounts of free GABA (0.35–0.41 mg/kg/day), our findings indicate that reduced macrophage infiltration in males may have helped counterbalance the colonic responses to regular IC (caffeine) intake, possibly through GABA-related pathways.

The levels of free GABA present in IC beverage are below the doses required to modulate locomotor activity or pain sensitivity in previous studies using oral administration in preclinical models [54]. Furthermore, its poor bioavailability through the oral route and its restricted capacity to cross the blood–brain barrier limits the systemic and central effects of orally administered GABA [57]. However, GABAergic neurotransmission could still be modulated indirectly, as melanoidins present in IC (estimated intake: 155 mg/kg/day in males and 189 mg/kg/day in females) are known to stimulate GABA-producing bacteria [58,59]. In a previous study, melanoidins also reduced oxidative stress and improved intestinal transit in male rats (female rats were not studied), maybe through SCFA production [10], and SCFAs influencing the activity of GABA-producing bacteria can also be produced from polysaccharides, present in IC, (estimated intake: 189 mg/kg/day in males and 231 mg/kg/day in females), that can be fermented by gut microbiota [60]. Although no studies have directly linked IC-derived polysaccharides to the stimulation of GABA-producing microorganisms via SCFA production, related coffee fractions rich in arabinogalactans and melanoidins have shown this potential in vitro through modulation of microbiota and increased SCFA formation [61]. Despite the relatively low intake of IC melanoidins and polysaccharides, their potential role as modulators of the gut microbiota may influence the effects of other IC bioactive compounds, such as caffeine, CGA, and GABA. Furthermore, CGA may indirectly contribute to GABA synthesis by modulating the activity of glutamate decarboxylase, the key enzyme responsible for GABA production, potentially enhancing its local levels in the gut [62]. Indeed, in IC, GABA was detected along with glutamic acid (Glu) and aspartic acid (Asp) (Appendix A), both key precursors for in vivo GABA synthesis [63]. Glu can be directly converted into GABA via glutamate decarboxylase, while Asp contributes indirectly to the Glu pool through transamination reactions, providing additional substrate for GABA production [63]. Therefore, the contribution of IC to GABA availability in the body encompasses both the naturally occurring GABA and that which may be formed from Glu, Asp, and microbiota-mediated pathways [63].

Clearly, IC components may interact, causing effects that extend beyond their individual actions. Indeed, this study assessed IC as a whole beverage matrix rather than as isolated components. This approach is inherently complex, since bioactive compounds may interact in synergistic or antagonistic ways, and the amounts that ultimately reach target tissues are influenced by digestion, metabolism, and, to some extent, microbial transformation. These challenges, however, make the strategy more physiologically relevant, as it more closely reflects real-life consumption than single-compound studies. Accordingly, the effects observed here should be viewed as the outcome of multiple interacting processes, underscoring both the difficulty and the added value of investigating IC as a complete food matrix.

This study has some limitations. The use of Wistar rats limits translational relevance, underscoring the need for confirmation in other models and humans. Although no differences were detected between female groups when estrous cycle phases were analyzed in two categories, it cannot be excluded that a more detailed four-phase distribution contributed to some of the effects observed in females. Moreover, only colonic markers were analyzed, leaving out potential contributions from other tissues. Microbiota composition, cytokine profiles, and systemic immune responses were not assessed, which restricts mechanistic interpretation. Clarifying the interactions among IC components within this matrix will be essential to understand its sex-dependent effects.

## 4. Materials and Methods

### 4.1. Ethics Statement

The experimental protocol was approved by the Ethic Committee of Rey Juan Carlos University (URJC) and Comunidad de Madrid (PROEX-059/2018) and complied with the European Community Council Directive of 22 September 2010 (2010/63/EU) and Spanish (Law 32/2007, RD 53/2013, and order ECC/566/2015) regulations, for the protection of animals intended for scientific studies.

Additionally, the ARRIVE guidelines (available at https://arriveguidelines.org/arrive-guidelines; accessed on 23 May 2025) were strictly followed. Accordingly, the animal research facility provided researchers with predefined severity criteria to assess the welfare of the animals. For example, if an animal shows signs of excessive distress, including more than a 10% loss in body weight, euthanasia is performed to prevent further suffering. The study prioritized reducing pain and discomfort, as well as limiting the number of animals used to the minimum required.

### 4.2. Animals and Experimental Groups

Young adult Wistar HAN healthy rats (3 months old) from the URJC Veterinary Unit were used. The study included 9 female rats (200–250 g) and 9 male rats (250–300 g) in cohort 1, and 6–9 female and male rats with the same age and weight ranges in cohort 2. Animals were housed in standard transparent cages (60 cm × 40 cm × 20 cm), 3–4 animals per cage, separated by sex, at a constant temperature (22 ± 0.5 °C) and relative humidity (55 ± 3%). In addition, 12/12 h light/dark cycles were used, with lights on from 8:00 to 20:00. The animals had free access to chow pellets (SAFE D40 diet, www.safe-diets.com, accessed on 1 May 2025) and beverage (sterile tap water or IC).

In each cohort, rats were randomly allocated to four experimental groups based on sex and beverage (potential confounders were not controlled). Half of the male and female animals received sterile water in their drinking bottles, while the rest were exposed to IC beverage (10 mg/mL in sterile water).

### 4.3. Instant Cascara Beverage

Coffee cascara from Arabica species and Tabi variety from Colombia, was provided by SUPRACAFÉ S.A. (Madrid, Spain). The method for obtaining the powdered coffee cascara extract is described in patent WO2013004873A [64]. The method consists of an aqueous extraction of 50 g/L at 100 °C for 10 min (20% extraction yield), and afterwards the sample was filtered (250 μm) and freeze-dried. IC powder was added to water at a concentration of 10 mg/mL in the bottle, and the resulting mixture was thoroughly homogenized to ensure a consistent composition. This homogenized solution constitutes what is commonly referred to as IC beverage. This concentration (10 mg/mL) was chosen based on previous studies [23,24]. A detailed description of the nutritional composition of IC beverage (10 mg/mL), including macronutrients, minerals, and bioactive compounds, is available as Appendix A.

### 4.4. Experimental Protocol

Animals were exposed to water (control group) or IC beverage for three consecutive weeks, and different parameters were evaluated (Figure 5).

In Cohort 1, body weight, food, and liquid intake were regularly monitored throughout the study. Somatic sensitivity and locomotor activity were evaluated during the second week after initiating beverage exposure, and during the third week, visceral sensitivity was assessed through a colorectal distension study. Following this assessment, the animals were humanely sacrificed using an intraperitoneal overdose of pentobarbital (60–80 mg/kg), followed by sharp decapitation.

In Cohort 2, histological and immunohistochemical studies were performed on colonic tissue samples collected from animals evaluated in a previous study, in which the same duration of IC exposure was applied [23]. Samples from Cohort 1 were not used for this to avoid possible alterations in the studied parameters associated with the colorectal distension test used for the study of visceral pain.

Females from both cohorts underwent vaginal cytology to determine the phase of the estrous cycle (see Section 4.6).

All experimental procedures were performed and analyzed by a researcher blind to the treatments the animals received.

### 4.5. Body Weight, Solid and Liquid Intake

During the three weeks of IC or water exposure, body weight and food and liquid intake were manually measured three days a week. The water was changed twice a week, and the IC beverage was changed three times a week, with any leftovers being discarded.

Body weight is represented as the weight gained throughout the three weeks of the study. Food and liquid intakes are represented as the mean intake per rat and day for each experimental group, with measurements initially taken per cage and then divided by the number of rats per cage.

### 4.6. Vaginal Cytology Smear in Female Rats

Vaginal smears were taken from female subjects at different times: after testing for locomotor activity, somatic and visceral sensitivity, and prior to sacrifice. Using a cotton-tipped applicator rotated three times, samples were collected 2 cm from the vaginal opening. These smears, mounted on slides and stained with hematoxylin-eosin (H&E), were microscopically examined to determine the estrous cycle phase [23,26]. The rat estrous cycle includes proestrus, estrus, metestrus, and diestrus phases, each characterized by specific cell types and correlating with hormonal changes in the human menstrual cycle [65]. For analytical purposes, the phases were grouped into two categories, proestrus/estrus and metestrus/diestrus, based on their similar hormonal profiles, to simplify statistical comparisons and enhance the interpretability of the results [66].

### 4.7. Locomotor Activity Analysis

At the end of the second week, after the assessment of somatic sensitivity, spontaneous locomotor activity was analyzed in cohort 1 using individual photocell activity chambers, in accordance with previously described methods [25]. Thus, each rat was individually placed into a distinct recording chamber (Cibertec S.A, Madrid, Spain; dimensions: 55 cm × 40 cm; beam spacing: 3 cm), and the number of interruptions of photocell beams were recorded for 30 min. The average count of crossings over the photocell beams for 3 consecutive periods of 10 min within those 30 min was employed for comparative analysis.

### 4.8. Somatic Sensitivity

#### 4.8.1. Von Frey Test

The Von Frey test was performed to assess mechanical sensitivity to a non-painful tactile mechanical stimulus. A series of calibrated Von Frey filaments (6–26 g; Ugo Basile) were used to measure the withdrawal threshold. Rats were individually accustomed to an elevated iron mesh covered by a transparent cage (10 cm × 20 cm × 15 cm) for 15 min. Habituation to this environment was also performed two days before assessment. The nociceptive test was carried out by applying each Von Frey filament to the hind paws of the animals through the mesh. The process was repeated 5 times per hind paw with a stimulation interval of 30 s. The result was considered positive when three of the five attempts induced a withdrawal response with the same filament. The mechanical threshold was determined with the lowest filament that was considered to induce a positive response, and the average of the results obtained for both paws was used for data analysis [25].

#### 4.8.2. Pressure Administration Device Test

To ascertain the threshold of sensitivity of the skeletal muscle to unpleasant mechanical stimuli, the pressure administration (PAM) device was employed on the gastrocnemius muscle, following established protocols [67]. This device consists of a force transducer mounted on a unit that fits onto the operator’s thumb and has an eight mm diameter circular contact. The rat was carefully held by the researcher, wrapped in a cloth, leaving the hindlimb accessible; the rat was accustomed to this handling for two days before starting the experiments. The force applied to the gastrocnemius muscle was increased steadily at a rate of 50 g per second until the rat either withdrew its hindlimb, vocalized, or a maximum pressure of 400 g-force (g/f) was achieved. The peak force applied just before hindlimb withdrawal or vocalization was recorded, and the average of three consecutive measurements was taken as the nociceptive mechanical threshold.

### 4.9. Visceral Sensitivity

During the third week of the study, visceral pain was assessed in cohort 1. Following sedation with dexmedetomidine (Sedator^®^, 1 mL/kg, 1 mg/mL; intraperitoneal), a longitudinal line measuring 10 cm along the linea alba of the rat abdominal region and transverse lines at 2 cm intervals were drawn [26]. Rectal contents were gently removed, and a 5 cm latex balloon coated with Vaseline was introduced into the colon, positioning its tip 7 cm within the colorectum. The sedation effect was counteracted by administering atipamezole (Revertor^®^, 0.66 mL/kg, 5 mg/mL; intraperitoneal), and the behavior of the rat was recorded for a duration of 35 min using a video camera (iPad, Apple Inc., Cupertino, CA, USA) situated 30 cm below the floor of the cage. After an initial 5 min interval, which was not used for the analysis, the pressure within the intracolonic balloon was progressively elevated by means of a sphygmomanometer (Riester GmbH, Jungingen, Germany). For this, the pressure was raised from 0 to 80 mm Hg, in increments of 20 mm Hg every 5 min, before ultimately being reset to 0 mm Hg. A phasic stimulation protocol was applied in accordance with prior descriptions [26], for which a uniform stimulus lasting 20 s was reiterated thrice within each 5 min period, interspersed by a stimulus-free interval of at least 1 min. The captured videos were converted into a sequence of individual frames (at a rate of 1 frame per second) using QuickTime Player Pro for Windows (version 7.7.4; Apple Inc., Cupertino, CA, USA). Using these frames, the mean count and duration of contractions, along with the proportion of time during which the rat abdomen exhibited contractions, were assessed for each distinct pressure stimulus.

### 4.10. Enteric Glial Cells and CGRP-IR Varicose Fibers in the Distal Colon Myenteric Plexus

Distal colon samples (2 cm segments) obtained from cohort 2 were placed in Krebs’ solution (118 mM NaCl, 4.75 mM KCl, 1.2 mM MgSO_4_, 1.19 mM KH_2_PO_4_, 2.54 mM CaCl_2_, 25 mM NaHCO_3_, 11 mM glucose; pH 7.4) to obtain whole-mount preparations of the myenteric plexus. The samples were then stretched and pinned onto a Petri dish coated with Sylgard^®^ (The Dow Chemical Company, Midland, MI, USA) and filled with Krebs solution, previously aerated with carbogen (95% O_2_, 5% CO_2_). The mucosal and submucosal layers were manually dissected using fine forceps. Then, the tissue was immersed in Zamboni’s fixative (15% picric acid, 20% paraformaldehyde, dissolved in phosphate buffer) for 24–48 h at 4 °C and subsequently permeabilized using dimethyl sulfoxide (3 × 10 min) followed by phosphate-buffered saline (PBS) (3 × 10 min). Finally, the circular smooth muscle layer was removed, leaving only the longitudinal muscle layer with the myenteric plexus attached. The whole-mount preparations obtained were conserved at 4 °C in PBS with sodium azide (1%), until processing [24,68].

Immunohistochemistry methods were performed to detect structures of interest within the myenteric plexus. For these, tissues were incubated 3 days (24 h at room temperature and, the remaining 48 h, at 4 °C) with the selected primary antibodies. For the detection of enteric glial cells, the antibody against the transcription factor Sox-10 was used (mouse anti-Sox-10 (A2), 1:500; sc-365692, Santa Cruz Technologies, Dallas, TX, USA). In addition, an antibody against CGRP was used (rabbit anti-CGRP, 1:500; ab47027, Abcam, Cambridge, UK). After washing with PBS (3 × 10 min), tissues were exposed for 24 h (at room temperature) to the secondary antibody: donkey anti-mouse RRX (1/500; 715-295-151, Jackson, West Grove, PA, USA) and donkey anti-rabbit CY5 (1/500; 711-175-152, Jackson). Finally, preparations were washed again with PBS (3 × 10 min), dehydrated in 50%–70%–100% buffered glycerol (10 min each) and mounted on slides. In addition, a negative control was included to discard nonspecific labeling. For this, tissue samples were incubated in the absence of the primary antibody, and the entire protocol was completed.

The preparations were analyzed under a fluorescence Nikon Eclipse microscope, equipped with a Nikon Moments camera with NIS-Elements BR 5.30.05 software (Nikon, Tokio, Japan). The analysis was performed on 6–9 whole-mount preparations per treatment, in 8–10 non-overlapping microphotographs (20× magnification) per preparation and marker using the program ImageJ Fiji (ImageJ, an open-source software developed by the National Institutes of Health, Bethesda, MD, USA, available at https://imagej.net/Fiji, accessed on 23 March 2025).

Regarding the enteric glia, the total glial density (number of glial cells per serosal surface unit), the packing density (number of intraganglionic enteric glial cells per ganglionic area), and the density of extraganglionic glial cells (per serosal surface area) were calculated. Also, the relationship between enteric glial cells immunoreactive (IR) for Sox-10 (Sox-10-IR) and the number of HuC/D-IR neurons was estimated. For this, HuC/D-IR neuronal density (per serosal surface area) was obtained from our previous work using similar whole-mount preparations [24], and the ratio of Sox-10-IR myenteric glial cells per myenteric neuron was obtained.

Finally, a semiquantitative analysis of the presence of varicose CGRP-IR fibers was performed using the following criteria: 0-non-existing, 1-low, 2-normal, 3-high [68].

### 4.11. Immune Cells in the Distal Colon Wall

#### 4.11.1. Mast Cell Quantification

Histological samples from the distal colon were collected from the animals in cohort 2, following the completion of the experiments previously performed at our laboratory [23]. After sacrifice, a 1 cm portion of the distal colon, was fixed in 10% formalin, embedded in paraffin, and sliced into 5 µm sections. These sections were then deparaffinized and rehydrated before being stained with toluidine blue for mast cell quantification. The evaluation was performed using a Zeiss Axioskop 2 microscope (Carl Zeiss AG, Oberkochen, Germany) equipped with the image analysis software package LAS X (Leica Microsystems GmbH, Wetzlar, Germany), counting samples under a ×40 objective in ten non-overlapping fields per colon sample, focusing on the area across mucosa and submucosa layers [25].

#### 4.11.2. Eosinophils

To assess eosinophil presence in colon samples, tissues were initially fixed in 10% formalin and subsequently processed for paraffin embedding. Thin histological sections (4–5 µm) were prepared, mounted on slides, and stained with H&E (hematoxylin and eosin). This method enabled the identification of eosinophils by their characteristic reddish eosinophilic cytoplasmic granules and bilobed nuclei. The analysis was performed using a Zeiss Axioskop 2 microscope equipped with the image analysis software package LAS X, allowing precise image acquisition and evaluation of eosinophilic infiltration specifically in the submucosa of the colon, across the experimental groups, using a x40 objective.

#### 4.11.3. Macrophages and Neutrophils

For immunohistochemistry, the colon samples were deparaffinated and rehydrated. Antigen unmasking was achieved by heating in a microwave oven at 98 °C in a 10 mmol/L citrate buffer for 30 min. Then, sections were treated with 3% (vol/vol) hydrogen peroxide for 10 min to inhibit endogenous peroxidase activity and blocked with normal horse serum (BioRad, Berkeley, CA, USA) for 20 min. Thereafter, sections were incubated overnight at 4 °C with mouse anti-CD163 antibody (clone ED2, BioRad; MCA342GA, 1/1000) to reveal M2 macrophages and rabbit anti-myeloperoxidase (MPO, ab65871, Abcam, 1/1000) to detect neutrophils. The ImmPRESS^®^ HRP Universal peroxidase kit (Vector Laboratories Inc., Burlingame, CA, USA) served as the secondary polymer. Hematoxylin was used for counterstaining, and the samples were mounted with Eukitt mounting media (O. Kindler GmbH & Co., Freiburg, Germany). Negative controls consisted of samples that were not treated with the primary antibody. Colon samples were analyzed using a Zeiss Axioskop 2 microscope, integrated with LAS X imaging software (version 3.0.5, Leica Microsystems, Wetzlar, Germany; available at https://www.leica-microsystems.com, accessed on 4 April 2025). Ten photographs per animal at ×20 or ×40 magnification were taken and assessed using Image J-Fiji to determine the number of immunocytes present in the colonic samples, located in the mucosa and submucosa [69].

### 4.12. Statistical Analysis

Sample size was determined a priori using G*Power for a mixed ANOVA (repeated measures, within–between interaction), assuming a medium effect size (f = 0.25), α = 0.05, power = 0.80, four between-subject groups (sex × treatment), and six repeated measurements. The analysis yielded a total of 28 animals (7 per group); in practice, 9 animals per group were included, equally distributed by sex, with some analyses limited to 6 animals due to data loss. The sample size was based on visceral pain, the most variable endpoint, and applied to other parameters to ensure consistent power. This design optimized statistical power for the mixed-model ANOVA, whereas one-way analyses performed on locomotor, somatic, or histological data had inherently lower sensitivity, which was considered when interpreting these outcomes.

All collected data were included in the analyses without exclusions. Each animal was treated as an independent experimental unit for body weight, locomotor activity, and pain-related measures, whereas the cage was considered the unit for food and water intakes. Immunohistochemical and immunological variables (CGRP-IR, Sox10-IR, and immune cell counts) were analyzed on a per-animal basis.

Visceral pain responses were analyzed using a mixed-model ANOVA with repeated measures, with pressure (0–80 mmHg) as a within-subject factor and sex and treatment as between-subject factors. For normally distributed data, one-way or two-way ANOVA was used, followed by Bonferroni’s or Tukey’s post hoc tests; non-normally distributed data were analyzed using the Kruskal–Wallis’ test followed by Dunn’s multiple comparison test. Fisher’s exact test was applied to estrous cycle distribution. Normality and homogeneity of variance were assessed using the Shapiro–Wilk’s test. GraphPad Prism 8.0.2 (La Jolla, CA, USA) was used in these analyses. Statistical significance was set at *p* < 0.05.

Symbolic representation in graphs employed # and $ to denote statistically significant differences linked to the sex of the animals (males vs. females; # without and $ with IC), while * and + symbols were utilized to signify variances attributed to beverage administration (control vs. IC beverage; * in males and + in females). The symbol & was used to indicate time-dependent changes in spontaneous locomotor activity.

## 5. Conclusions

Regular consumption of the IC beverage led to clear sex-specific effects in healthy rats. Females showed increased sensitivity to pain in a statistically significant manner for the skeletal muscle, although not for the colon, which may be due to opposite (compensatory) changes occurring in the enteric nervous (more enteric glial cells, but fewer CGRP-IR nerve fibers) and immune systems (sex-dependent differences in macrophage infiltration).

These effects may be due to the independent or combined action of several components in the IC beverage (like caffeine, CGA, GABA, melanoidins and polysaccharides), how they interact in the food matrix, and how they are transformed by both abiotic and biotic digestion into different metabolites with their own potential activities, both at the local and systemic levels, after absorption. Differences between males and females in hormone levels, metabolism, and how their bodies respond to these compounds may also play a role.

Altogether, these results highlight the importance of studying complex food matrices like IC, where bioactive compounds act synergistically or antagonistically, and point to the need for further studies to unravel the mechanisms underlying sex-dependent neuroimmune modulation of the BGA and pain sensitivity, as well as their translational relevance to human health.

## Figures and Tables

**Figure 1 ijms-26-10691-f001:**
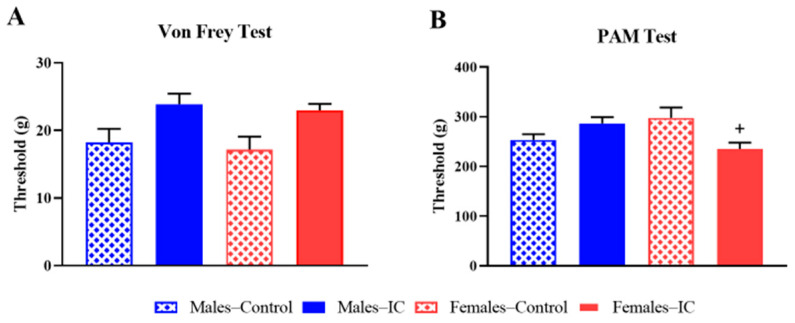
Somatic sensitivity. Nociceptive thresholds were evaluated using two complementary assays: the von Frey test (**A**) and the pressure application measurement (PAM) test (**B**). Data are expressed as mean ± SEM (*n* = 9/group) for 4 experimental groups defined by sex (Males or Females) and treatment (Control or IC). Significant differences due to IC intake are indicated as + *p* < 0.05 (Females–IC vs. Females–Control). Statistical analysis was performed using one-way ANOVA with Tukey’s post hoc test for von Frey data (**A**), and Kruskal–Wallis’ followed by Dunn’s multiple comparison test for PAM data (**B**).

**Figure 2 ijms-26-10691-f002:**
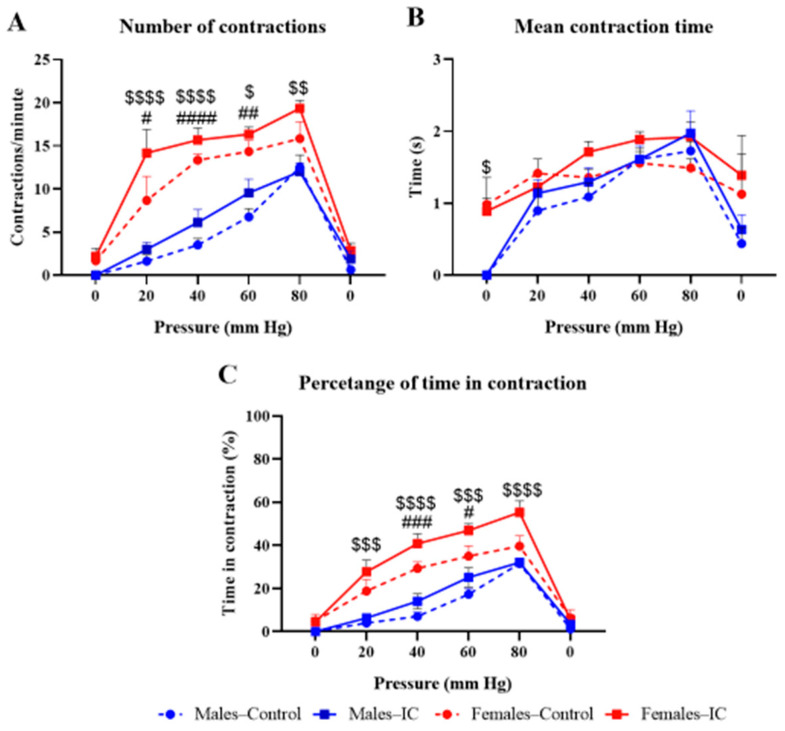
Visceral sensitivity. Abdominal contractions were assessed during intracolonic balloon distension under a phasic stimulation protocol in which distensions ranged from 0 to 80 mmHg, applied in 20 s pulses (3 per pressure level, with ≥60 s intervals). Three measures were obtained: number of contractions (**A**), mean contraction duration (**B**), and percentage of time in contraction (**C**). Data are expressed as mean ± SEM (*n* = 9/group) for 4 experimental groups defined by sex (Males or Females) and treatment (Control or IC). Significant sex effects were observed (# *p* < 0.05, ## *p* < 0.01, ### *p* < 0.001, #### *p* < 0.0001 for Females–Control vs. Males–Control; $ *p* < 0.05, $$ *p* < 0.01, $$$ *p* < 0.001, $$$$ *p* < 0.0001 for Females–IC vs. Males–IC), whereas IC administration did not significantly modify any parameter compared with control. Statistical analysis was performed using a mixed-model ANOVA with Tukey’s post hoc tests for normally distributed data.

**Figure 3 ijms-26-10691-f003:**
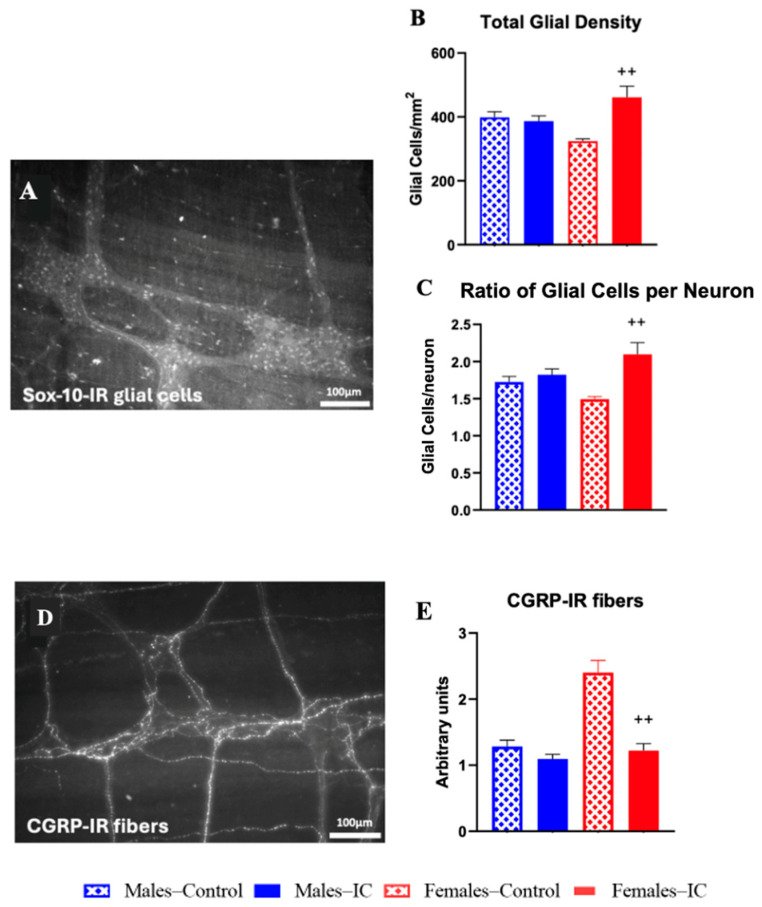
Enteric Glial Cells Immunoreactive (IR) to Sox-10 (Sox-10-IR) and Calcitonin Gene-Related Peptide (CGRP)-IR Varicose Fiber Analysis. (**A**) Representative Sox-10-immunoreactive (IR) glial cells located in the myenteric plexus of the distal colon (scale bar: 100 µm). (**B**) Quantification of Sox-10-IR glial cell density in the myenteric plexus. (**C**) Glia-to-neuron ratio calculated from Sox-10-IR glial cells and HuC/D-IR neurons. (**D**) Representative image of CGRP-IR varicose fibers in the myenteric plexus, visible both inside ganglia and along interganglionic fibers (scale bar: 100 µm). (**E**) Semiquantitative analysis of CGRP-IR fiber density in the distal colon myenteric plexus. Data are expressed as mean ± SEM (*n* = 6/group) for 4 experimental groups defined by sex (Males or Females) and treatment (Control or IC). Significant beverage-related differences are indicated as ++ *p* < 0.05 (Females–IC vs. Females–Control). Statistical tests: one-way ANOVA combined with Kruskal–Wallis’ and Dunn’s multiple comparisons tests.

**Figure 4 ijms-26-10691-f004:**
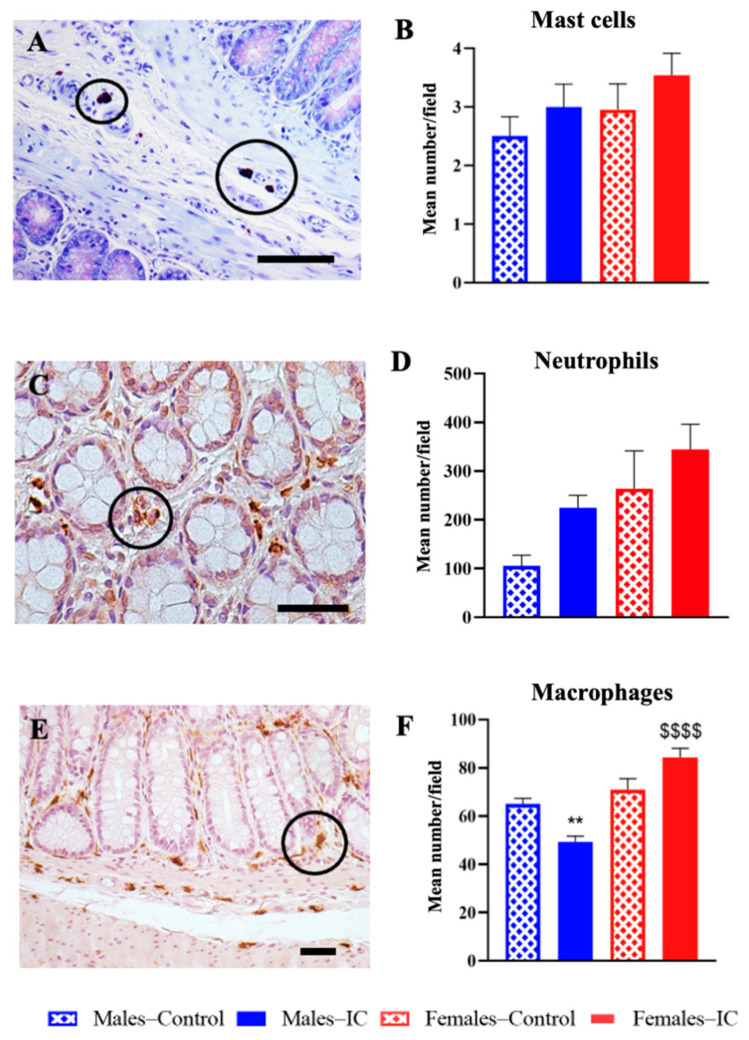
Immune Cell Quantification in Colon Sections. Representative images of (**A**) Mast cells, (**C**) neutrophils, and (**E**) macrophages in colon sections. Black circles highlight representative immune cells identified for each cell type. Mast cells were stained with toluidine blue, while neutrophils and macrophages were detected with specific antibodies. Quantification of cell density (**B**,**D**,**F**) was performed in ten randomly selected fields per section. Data are expressed as mean ± SEM (*n* = 6/group) for 4 experimental groups defined by sex (Males or Females) and treatment (Control or IC). Statistically significant differences were observed in macrophage counts: sex-related differences ($$$$ *p* < 0.0001, Females–IC vs. Males–IC) and beverage-related differences (** *p* < 0.01, Males–IC vs. Males–Control). One-way ANOVA followed by Kruskal–Wallis’ and Dunn’s multiple comparison tests. Scale bar: 50 μm.

**Figure 5 ijms-26-10691-f005:**
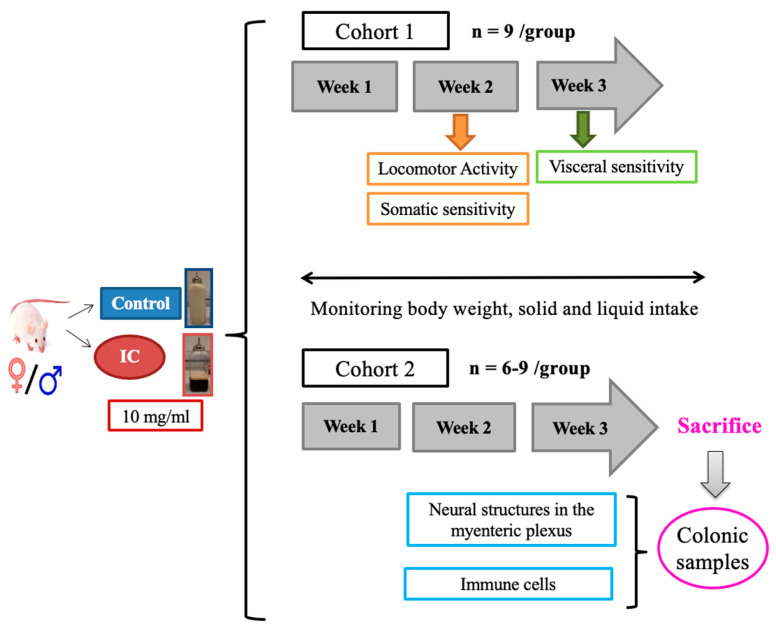
Experimental protocol. Adult, sexually mature male and female rats were exposed to water (control group) or Instant Cascara (IC, 10 mg/mL) beverage for 3 weeks. In Cohort 1, body weight, food, and liquid intake were monitored throughout the whole study, and locomotor activity and somatic sensitivity were evaluated during week 2, followed by an assessment of visceral sensitivity during week 3. After the visceral sensitivity study, rats were sacrificed, and no colonic samples were collected. Animals used in a previously published study [23], provided colon samples for histological and immunohistochemical studies (Cohort 2). Vaginal cytology was conducted in all females after performing the tests for locomotor activity, somatic and visceral sensitivity (cohort 1), and prior to sacrifice (cohort 2).

**Table 1 ijms-26-10691-t001:** Effect of regular exposure to Instant Cascara (IC) beverage on spontaneous locomotor activity.

Period of Time	Males–Control	Males–IC	Females–Control	Females–IC
0–10 min	629 ± 54	694 ± 59	598 ± 91	874 ± 64 **^+^**
11–20 min	308 ± 51 ^&^	387 ± 31 ^&&&&^	365 ± 55 ^&^	495 ± 55 ^&&&^
21–30 min	192 ± 72 ^&&^	135 ± 41 ^&&&&^	174 ± 46 ^&&&^	178 ± 62 ^&&&&^

An actimeter was used to record locomotor activity as the number of photobeam interruptions over 3 consecutive periods of 10 min each. Data are expressed as mean ± SEM (*n* = 9/group) for 4 experimental groups defined by sex (Males or Females) and treatment (Control or IC). Statistical analysis was performed using one-way ANOVA (for normally distributed data) or Kruskal–Wallis’ test (for non-normally distributed data). Temporal variation within the same group was assessed with Tukey’s post hoc test (^&^ *p* < 0.05, ^&&^ *p* < 0.01, ^&&&^ *p* < 0.001, ^&&&&^ *p* < 0.0001 vs. 0–10 min), while inter-group comparisons were assessed with Dunn’s post hoc test (**^+^** *p* < 0.05, Females–IC vs. Females–Control).

**Table 2 ijms-26-10691-t002:** Summary of Main Results.

Parameters		Changes by Sex	Changes by Beverage
General Health (body weight, solid and liquid intakes)		♀ < ♂	N.S.
Locomotion		N.S.	♀( ** IC ** > ** W ** )
Sensitivity	Tactile	N.S.	N.S.
	Skeletal muscle	N.S.	♀( ** IC ** > ** W ** )
	Visceral	♀ > ♂	N.S.
Colonic neuroimmune modulation	Glial cells	N.S.	♀( ** IC ** > ** W ** )
	CGRP varicose fibers	N.S.	♀( ** IC ** < ** W ** )
	M2 Macrophages	♀ > ♂ (only in **IC**)	♂ ( ** IC ** < ** W ** )

Effects of regular IC exposure in male (♂) and female (♀) rats, compared to exposure to water (control group). Abbreviations: IC, Instant Cascara; W, Water; CGRP, Calcitonin Gene-Related Peptide; N.S., Not Significant.

## Data Availability

The raw data supporting the conclusions of this article will be made available by the authors on request.

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
