# Peer review of "Instant Cascara Beverage as a Neuroimmune Modulator of the Brain–Gut Axis: Sex-Dependent Effects in Healthy Rats"

_ijms, 2025, doi:10.3390/ijms262110691_

Round 1

Reviewer 1 Report

Comments and Suggestions for Authors

Your manuscript is generally of high quality; however, I believe there are still some issues that need to be addressed in order to further improve the overall quality of the paper.

First, I find it difficult to grasp a comprehensive understanding of your experimental design and key findings from the abstract. You may need to reorganize and clarify the abstract to better reflect the core content of your study.

Second, from a professional perspective, your study is centered on a key compound, but the introduction focuses heavily on the gut-brain axis. I suggest you revise the introduction to better align with the research focus and refer to the structure of the introduction in the article “Linalool as a key component in strawberry volatile organic compounds (VOCs) modulates gut microbiota, systemic inflammation, and glucolipid metabolism.”

Third, the logical connections between the various indicators measured in your study are not clearly explained. As a reader, it is difficult to understand how these indicators relate to each other. I recommend that you consult and cite “β-ionone prevents dextran sulfate sodium-induced ulcerative colitis and modulates gut microbiota in mice” for guidance on how to establish clearer connections among your study parameters.

Fourth, in the Results and Discussion section, although your introduction mentions gut microbiota and metabolites, these aspects are not sufficiently addressed in the results. Additionally, you focus extensively on the role of caffeine, which is not the same compound as the main substance under study. You should either explain why caffeine is emphasized or provide experimental evidence that caffeine is the primary active compound—similar to the approach used in the previously mentioned reference articles.

Furthermore, you later introduce inflammation-related content, yet there are no corresponding results presented. Many of your interpretations seem to be speculative. Your indicators are mostly focused at the compound level, lacking data on downstream regulatory substances. Please take this issue seriously and consider including relevant detection results.

Finally, as you stated in the manuscript, there is still much to be explored. However, based on the current experimental data, you should be able to draw some definitive conclusions. At present, the conclusion section merely summarizes the results without discussing the underlying mechanisms or causes. The major conclusions derived in the discussion should be clearly stated in the conclusion section to serve as a strong support for the manuscript.

Reviewer 2 Report

Comments and Suggestions for Authors

This study provides novel insights into the sex-specific effects of Instant Cascara (IC) on neuroimmune interactions and pain sensitivity in rats. The experimental design is generally robust, and the findings highlight significant sex-dependent outcomes relevant to Instant Cascara beverage research. However, there are still some issues that need to be addressed. Specific comments

Abstract

  1. L24-26 The term "sex-specific neuroimmune mechanisms" requires further clarification, as the current research has not fully established the causal relationship of these mechanisms.
  2. L30-31 Visceral sensitivity results: Specify significance occurred only at 80 mmHg (not across all pressures).

Introduction

  1. Modify L 80 Coffea arabica to italics.
  2. The introduction of the beneficial effects of Cascara beverage is not deep enough. Please provide an in-depth introduction to the mechanism.
  3. The existing literature does not sufficiently address the mechanisms underlying gender differences. Please provide further clarification or supporting information.

Results

  1. Sample size justification: State that power analysis was performed for all endpoints (not only visceral pain).

Discussion

  1. In males, reduced M2 macrophages (Fig 4F) lacked corresponding visceral hypersensitivity. Discuss potential compensatory mechanisms.
  2. Address possible synergistic/antagonistic effects of other IC components (e.g., GABA, melanoidins) in the observed sex effects. Balanced discussion of all bioactive components (not just caffeine).

  1. Please thoroughly check the format of the references.

Reviewer 3 Report

Comments and Suggestions for Authors

This manuscript presents an original and well-designed study investigating the impact of regular consumption of an Instant Cascara (IC) beverage on somatic and visceral sensitivity in rats, with a focus on sex-dependent differences. The study integrates behavioral assays, locomotor activity assessment, and immunohistochemical analyses to explore neuroimmune mechanisms potentially involved in pain modulation.

The work is relevant to the fields of nutrition, neurobiology, and gut-brain axis research. The findings contribute valuable insights into sex-specific physiological responses to dietary bioactives, particularly from coffee by-products such as cascara.

Major comments

  1. The introduction is well-structured and provides sufficient background on pain sensitivity, neuroimmune interactions, and the potential bioactive compounds in cascara. However, the authors should briefly discuss any prior research on cascara or similar compounds in pain modulation to strengthen the background and better highlight the novelty of their study.
  2. The methodology is rigorous, using validated behavioral tests (Von Frey, PAM, colorectal distension), locomotor activity measures, and immunohistochemical analysis. Inclusion of both male and female rats, along with vaginal cytology to monitor the estrous cycle, is a significant strength. Some points need to be improved:
    1. Clarify whether the sample size per group was determined based on power analysis.
    2. Although estrous cycle monitoring was performed, the authors might discuss whether any intra-group variation in females was attributable to specific phases.
    3. The potential role of gut microbiota is acknowledged but not directly addressed. The absence of microbiota profiling should be stated more explicitly as a limitation.
  3. Results are presented clearly with appropriate statistical analysis and visualization. Sex-specific responses are well-emphasized, particularly the neuroimmune changes observed in female rats. Consider some suggestions to implement:
    1. Include a consolidated table summarizing key findings across behavioral and immunological parameters for better readability.
    2. Briefly discuss statistical power and effect sizes to enhance confidence in the conclusions.
  4. The conclusions are well-supported by the results and appropriately cautious in their interpretations. However, the authors should explicitly state the absence of gut microbiota data and systemic inflammatory markers as limitations. Suggest concrete directions for future research, such as microbiota profiling, systemic cytokine measurement, or use of additional animal models to confirm translational potential.

Minor comments

  1. The manuscript is generally well-written with clear language, but a minor grammatical review could further improve clarity.
  2. Ensure that all abbreviations are defined at first use in the text and figures.

Recommendation

Minor Revision

This is a valuable and well performed study that advances understanding of sex-dependent neuroimmune responses to dietary bioactives. The revisions needed are primarily related to contextual discussion and explicit acknowledgment of study limitations. After addressing these minor points, the manuscript will be suitable for publication in IJMS.
